# On Averaging ROC Curves

**Jack Hogan**                                                                    *jack.hogan15@imperial.ac.uk*
*Department of Mathematics*
*Imperial College London, London, UK*

**Niall M. Adams**                                                                *n.adams@imperial.ac.uk*
*Department of Mathematics*
*Imperial College London, London, UK*

**Reviewed on OpenReview:** *https://openreview.net/forum?id=FByH3qL87G*

## Abstract

Receiver operating characteristic (ROC) curves are a popular method of summarising the performance of classifiers. The ROC curve describes the separability of the distributions of predictions from a two-class classifier. There are a variety of situations in which an analyst seeks to aggregate multiple ROC curves into a single representative example. A number of methods of doing so are available; however, there is a degree of subtlety that is often overlooked when selecting the appropriate one. An important component of this relates to the interpretation of the decision process for which the classifier will be used. This paper summarises a number of methods of aggregation and carefully delineates the interpretations of each in order to inform their correct usage. A toy example is provided that highlights how an injudicious choice of aggregation method can lead to erroneous conclusions.

## 1 Introduction

ROC curves are widely used to present performance assessment results in classification studies. Their usage spans many disciplines, including pattern recognition (Webb, 2003), medical diagnostics (Swets et al., 2000), consumer credit scoring (Hand & Henley, 1997), and biometrics (Ross & Jain, 2003). Typically, the design and construction of a classifier is an iterative process: at each stage, choices are made regarding feature selection, distributional assumptions etc., and the classifier must be re-evaluated to assess the impact of these choices on performance. In many classification problems, such as those with imbalanced classes or asymmetric misclassification costs, a scalar measure of performance such as accuracy is not sufficient (Provost & Fawcett, 1997). In addition, it is often the case that the misclassification costs and class frequencies are not known at experimentation time and will only become clear at the time of deployment (Drummond & Holte, 2000; Hernández-Orallo et al., 2012). ROC curves offer a useful visual evaluation of the trade-off between different types of error over the full range of possible costs and class frequencies. By examining a ROC curve, various operating points of interest—motivated by the specific application of the researcher—can be compared. If the operating conditions at deployment time are known precisely, then a plot of the entire ROC curve is redundant, as algorithms can be compared directly based on specific cost-sensitive classification metrics.

There are many situations in which there is a requirement to produce a *single* summary ROC curve from a collection of individual curves. For example, in credit card fraud detection, a bank may be running classifiers based on the same features for each customer (Juszczak et al., 2008); in biometric face verification, separate classifiers are trained for each individual subject (Marcialis & Roli, 2002); in radiology, multi-reader multi-case (MRMC) studies involve multiple radiologists producing scores for a collection of image cases (Skaron et al., 2012). Another situation involves the use of $k$-fold cross-validation procedures, which attempt to provide some sense of the average and the uncertainty in the ROC curve. It is perhaps natural in these settings to seek to report a single ROC curve summarising the performance of all the individual classifiers.

Such a curve will be useful for classifier development and selection. Note that a related problem is that of constructing a *covariate-adjusted* ROC curve (Janes & Pepe, 2009), which aims to account for factors or characteristics that influence the predictions made by a classifier on certain instances within a single dataset. The focus of this paper is on constructing a single ROC curve from *multiple* datasets.

To help reinforce the message of this paper, we provide three hypothetical examples of different situations in which ROC curves may be averaged:

- **Comparing classifiers:** The cyber-security team in an enterprise are testing two different algorithms for anomaly detection. For each device within the IT infrastructure, the algorithms model its network traffic and produce a probabilistic score indicating the degree to which it predicts an observation to be anomalous. Assume the number of observations varies between devices, based on their level of activity. Assume also that the true labels are known for at least some portion of the data, having been provided by security experts. A set of ROC curves can be constructed for both algorithms, based on their predictions on each device. The team are interested in comparing the average performance.

- **Comparing datasets:** A medical research team have developed a new type of imaging device for tumour detection. Images are produced using both the new and old device for a sample of patients whose tumour status is known. These images are presented to a number of radiologists (read classifiers), who produce independent sets of scores for each image. For each radiologist, two ROC curves can be constructed. The two imaging devices producing the input data can be compared by interpreting the average ROC curves across all radiologists.

- **Comparing hyperparameters:** A subscription business is developing a supervised learning model to predict churn; i.e. to identify subscribers that are likely to stop paying for the service. If churn can be accurately predicted, the business can pre-emptively contact the customer and offer them a discounted subscription in the hope of retaining their custom. To perform hyperparameter selection on their supervised learning model, they carry out 10-fold cross-validation using a range of possible values. For each value of the hyperparameter, an average ROC curve is used to assess and compare the performance of the classifier across the 10 folds of cross-validation.

Reasoning about a summary or "average" ROC curve in any of these contexts calls for fundamental considerations relating to the operational use of the classifier. Crucially, an aggregate ROC curve should represent performance in accordance with how the individual classifiers are to be deployed. As described in Section 2, a threshold $T$ is required to produce a decision from most classifiers. In these examples featuring multiple classifiers, the first question is whether every classifier is to employ the same threshold $T$ or if classifier $i$ operates using threshold $T_i$. Next, it is important that the method of aggregating individual ROC curves is compatible with the application-specific operating points or characteristics of interest, upon which performance assessments will be made. Once these considerations are made, it will be clear in Section 4.2, when we revisit the above examples, how an appropriate averaging should be performed.

The literature contains a number of methods for aggregating ROC curves (Swets & Pickett, 1982; Fawcett, 2006); however, clear guidance on their correct usage is lacking. Discussing two methods, *vertical averaging* and *pooling* (see Section 3), Witten et al. (2011) remark that "it is not clear which method is preferable. However, the latter method is easier to implement." Similarly, Parker et al. (2007) comment that "both strategies are used in practice and the current literature is equivocal about which approach is to be recommended." This is also clear when one considers software packages for ROC analysis; the R package `ROCR` offers three methods for averaging ROC curves, though neither its documentation (Sing et al., 2015) nor accompanying paper (Sing et al., 2005) provide any guidance on which is appropriate for different situations. An online vignette by the authors[1] demonstrates *threshold averaging* for combining the results of cross-validation. Meanwhile, a similar tutorial[2] for the Python package `scikit-learn` (Pedregosa et al., 2011) demonstrates *vertical averaging* in the same context.

---

[1] https://ipa-tys.github.io/ROCR/articles/ROCR.html
[2] https://scikit-learn.org/stable/auto_examples/model_selection/plot_roc_crossval.html

This paper aims to address this lack of guidance; we will show that care must be taken when choosing the appropriate averaging method, in particular with respect to the fundamental considerations described above. We will provide an example that demonstrates that incorrectly combining curves can lead to misrepresentation of results. The purpose of this paper is not to provide a deeply novel result but rather to clearly delineate the characteristics of different aggregation procedures and provide clear guidance for their correct usage.

The remainder of the paper proceeds as follows: Section 2 introduces the background and notation for individual ROC graphs. In Section 3 we describe and illustrate the construction of average ROC curves under the three proposed combination methods. In Section 4, a discussion of the interpretations of different aggregating methods motivates guidelines for selecting the appropriate method. Particular attention is paid to the hypothetical examples given above. Finally, Section 5 provides a simple illustration to demonstrate that the choice of combination method is crucial in the context of comparing classifiers.

## 2 ROC graphs

Here we follow closely the framework presented by Krzanowski & Hand (2009). Suppose that there exists two populations—a 'positive' population $P$ and a 'negative' population $N$—together with a classification rule for allocating unlabelled instances to one or other of these populations. A classification rule (*classifier*) is a function $S(\boldsymbol{X})$ of the random vector $\boldsymbol{X}$ of variables measured on each instance, used to predict class membership of a given instance. Some classifiers produce as output a discrete class label indicating only the predicted class of the instance; others produce a continuous output or *score*. The focus of this paper is the latter set of classifiers, as it is only for these that ROC curves can be produced. Suppose $\boldsymbol{x}$ is the observed value of $\boldsymbol{X}$ for a particular instance. The score $s(\boldsymbol{x})$ can be used to predict class membership according to whether $s(\boldsymbol{x})$ exceeds or does not exceed some threshold $T$. As $S$ is a continuous variable, we can consider the distribution of scores pertaining to instances from populations $P$ and $N$. Let $f_{S|P}(s) = p(s \mid P)$ and $f_{S|N}(s) = p(s \mid N)$ be the probability density functions of scores in $P$ and $N$, respectively, with corresponding distribution functions $F_{S|P}$ and $F_{S|N}$. A ROC graph is essentially a visual description of the degree of separability of these score distributions generated by a particular classifier.

We do not discuss how a classifier should be chosen or indeed calibrated for a particular task. These are well studied problems (Wolpert, 1996; Vapnik, 2013). Rather, we discuss how ROC curves are used to assess the performance of classifiers, in particular when applied to multiple datasets.

### 2.1 Definition

The score produced by a classifier can be a probability, representing the likelihood that an instance is a member of a particular class (the positive class, by convention), or simply a general, uncalibrated score, in which case it is conventional that higher scores are more indicative of positive class membership and lower scores indicative of negative class membership. In either case, the choice of threshold $T$ will dictate the class assignments made by the classifier. Suppose that $t$ is the value of the threshold $T$ chosen for a particular classifier; an instance is assigned to $P$ if its score $s$ exceeds $t$, otherwise $N$. In order to assess the efficacy of this choice of $T$, we must consider four possible outcomes and the *rate* at which these occur:

1. an instance from $P$ is correctly classified, i.e. the true positive rate $tp = p(s > t \mid P)$;

2. an instance from $N$ is misclassified, i.e. the false positive rate $fp = p(s > t \mid N)$;

3. an instance from $N$ is correctly classified, i.e. the true negative rate $tn = p(s \leq t \mid N)$;

4. an instance from $P$ is misclassified, i.e. the false negative rate $fn = p(s \leq t \mid P)$.

The choice of operating threshold $t$ is application-specific and often complicated. By varying $t$ and evaluating the four quantities above, we can inform a decision regarding which value to choose. In fact, as $tp + fn = 1$ and $fp + tn = 1$, we need only consider two of the above quantities, typically $fp$ and $tp$. A ROC curve is

obtained by varying $t$ from $-\infty$ to $\infty$ and tracing a curve of $(fp, tp)$. We can write down the equation of this curve in terms of the distribution functions defined above:

$$
\begin{aligned}
tp &= p(s > t \mid P), \quad -\infty < t < \infty \\
&= 1 - F_{S|P}(t), \quad -\infty < t < \infty \\
&= 1 - F_{S|P}\left[ F_{S|N}^{-1}(1 - fp) \right], \quad 0 \le fp \le 1.
\end{aligned}
$$

### 2.2 Estimation

The role of the ROC curve is to present an assessment of the performance of a classifier over the whole range of potential thresholds. This performance will be determined by the degree of overlap of the scores assigned by the classifier to instances from $P$ and $N$. In practice, we hardly ever know anything about the true underlying distributions of these scores. One typically only has available the values of $\boldsymbol{X}$ for a set of instances whose class labels are known. The data is commonly split into two portions: a *training set* is used to estimate any parameters required by the chosen classifier, and this classifier can then be applied to a *test set* in order to estimate the score distributions, and hence $fp$ and $tp$.

Standard methods of statistical inference are available to the researcher for this task. One may wish to assume parametric models for $F_{S|P}$ and $F_{S|N}$ and estimate parameters using maximum likelihood on the sample data. However, in this case, caution must be taken as the accuracy of the resulting ROC curve and any derived quantities strongly depends on the validity of the assumptions made (Krzanowski & Hand, 2009; Zhou et al., 2009). If a large number of test instances are available, then empirical estimation is preferred.

Let $n_P$ and $n_N$ be the number of instances in the samples from populations $P$ and $N$, respectively. We write $n_{P(t)}$ and $n_{N(t)}$ for the number of those instances whose classification scores are greater than $t$. Then the empirical estimators of $tp$ and $fp$ at the classifier threshold $t$ are given by

$$
\widehat{tp} = \frac{n_{P(t)}}{n_P} \qquad \text{and} \qquad \widehat{fp} = \frac{n_{N(t)}}{n_N}.
$$

The empirical ROC curve is then constructed by plotting the points $(\widehat{fp}, \widehat{tp})$ for varying $t$.

### 2.3 Area under the ROC curve

A quantity widely used to summarise an important element of the information portrayed by a ROC curve is the area under the curve (AUC). The AUC can be interpreted as an average true positive rate, regarding all values of the false positive rate as equally likely (Hand, 2009). Alternatively, it can be seen as the probability that the classifier will allocate a higher score to a randomly chosen instance from $P$ than it will to a randomly and independently chosen instance from $N$. Clearly, a higher AUC is desirable; however, classifiers should not be compared solely on their AUCs, as a higher AUC does not imply an everywhere dominating ROC curve. As a scalar measure of performance, a lot of valuable information contained in the ROC curve is lost, such as performance in specific regions of ROC space that may be of interest to the researcher. Moreover, there exists a fundamental incoherence in the use of AUC to compare *different* classifiers—see Hand (2009) for details, and Ferri et al. (2011) for an alternative interpretation of AUC.

## 3 Averaging ROC curves

The ROC graph described in the preceding section relates to a single dataset comprising instances belonging to either of two classes. Often we have multiple independent datasets of this type and a set of classification scores for each. In these settings, a separate ROC curve can be constructed from each set of scores. One approach for comparing different classifiers, or different iterations during classifier design and development, would be to compare average AUCs. Equally, we could compare the average accuracy at some known operating point. However, as discussed in Section 1, it is often the case that full knowledge of the operating conditions at deployment time is not available during experimentation. This is precisely why ROC curves are useful, and hence why we may want to construct a single ROC curve that serves as a summary or average

of the collection of individual ROC curves. In this way, the average performance at specific operating points can be assessed.

Methods for combining or averaging ROC curves have been proposed; however, the properties of the resulting curves have not been properly explored nor compared in this context. Moreover, there is very little guidance in the literature regarding which of the methods is most appropriate for a given task; in fact, existing guidance can even be misleading. As we will see, the interpretation of a summary ROC curve differs according to the averaging technique used, and so care should be taken both in choosing the appropriate method and in presenting results. Note that methods for averaging parametric ROC curves have also been proposed, based on averaging the estimated parameters of each curve (Metz, 1989). However, the focus of this discussion is on empirical ROC curves.

In this section we adapt the notation slightly to allow for the case where where there is a collection of datasets, each consisting of instances to be classified. Say that there are $M$ datasets, and dataset $i$ comprises a positive population $P_i$ and a negative population $N_i$. A classifier produces scores for each instance, so the ROC curve for dataset $i$ describes the separability of the score distributions $p(s \mid P_i)$ and $p(s \mid N_i)$.

For a simple illustration of the proposed averaging methods, we simulate scores arising from a probabilistic classifier employed on *two* datasets: the first 'well separable', the second 'poorly separable'. We assume equally balanced classes for both datasets but allow the total number of instances to differ between datasets. The number of scores simulated for the negative and positive classes of the well separable dataset was $n_{N_1} = n_{P_1} = 150$; for the poorly separable dataset $n_{N_2} = n_{P_2} = 75$. Density estimates of the simulated positive and negative scores for the two datasets are shown in the top two panels of Figure 1a.

### 3.1   Pooling

Swets & Pickett (1982) propose simply merging or *pooling* all the scores assigned to all instances from all datasets. A ROC curve is then constructed in the usual way as if every instance came from the one dataset:

$$\widetilde{tp} = \frac{\sum_{i=1}^{M} n_{P_i(t)}}{\sum_{i=1}^{M} n_{P_i}} \qquad\qquad \widetilde{fp} = \frac{\sum_{i=1}^{M} n_{N_i(t)}}{\sum_{i=1}^{M} n_{N_i}}$$
$$= \frac{n_{P(t)}}{n_P} \qquad\qquad\qquad = \frac{n_{N(t)}}{n_N},$$

where $P$ and $N$ are the total number of positive and negative instances among all $M$ datasets, respectively.

The pooled ROC curve is a representation of the degree of separability of the mixture distributions that result from pooling scores between the two datasets. Density estimates of these mixture distributions can be seen in the bottom panel of Figure 1a. A fixed threshold is shown in all three plots of Figure 1a and the corresponding points in ROC space can be seen on the ROC curves in Figure 1b.

### 3.2   Vertical Averaging

In Provost et al. (1998), *vertical averaging* is proposed, whereby points are sampled uniformly along the *fp* axis and the corresponding *tp* values from each individual ROC curve are averaged. Each ROC curve is treated as a function $R_i$ such that $tp = R_i(fp)$. This is done by choosing the maximum *tp* for each *fp* sampled between 0 and 1, interpolating between points where necessary. The vertical average ROC curve is

$$\overline{R}(fp) = \frac{1}{M} \sum_{i=1}^{M} R_i(fp), \quad 0 \leq fp \leq 1.$$

Standard errors can also be calculated and used to construct confidence intervals for the average curve (a related but different problem is that of constructing confidence *bands* for a ROC curve estimated from a single dataset—see Macskassy & Provost (2004)).

Note that the threshold that yields a given value of *fp* on each ROC curve will usually be different. The fixed threshold in Figure 1a yields a false positive rate of 0.12 for the well-separable dataset; the threshold

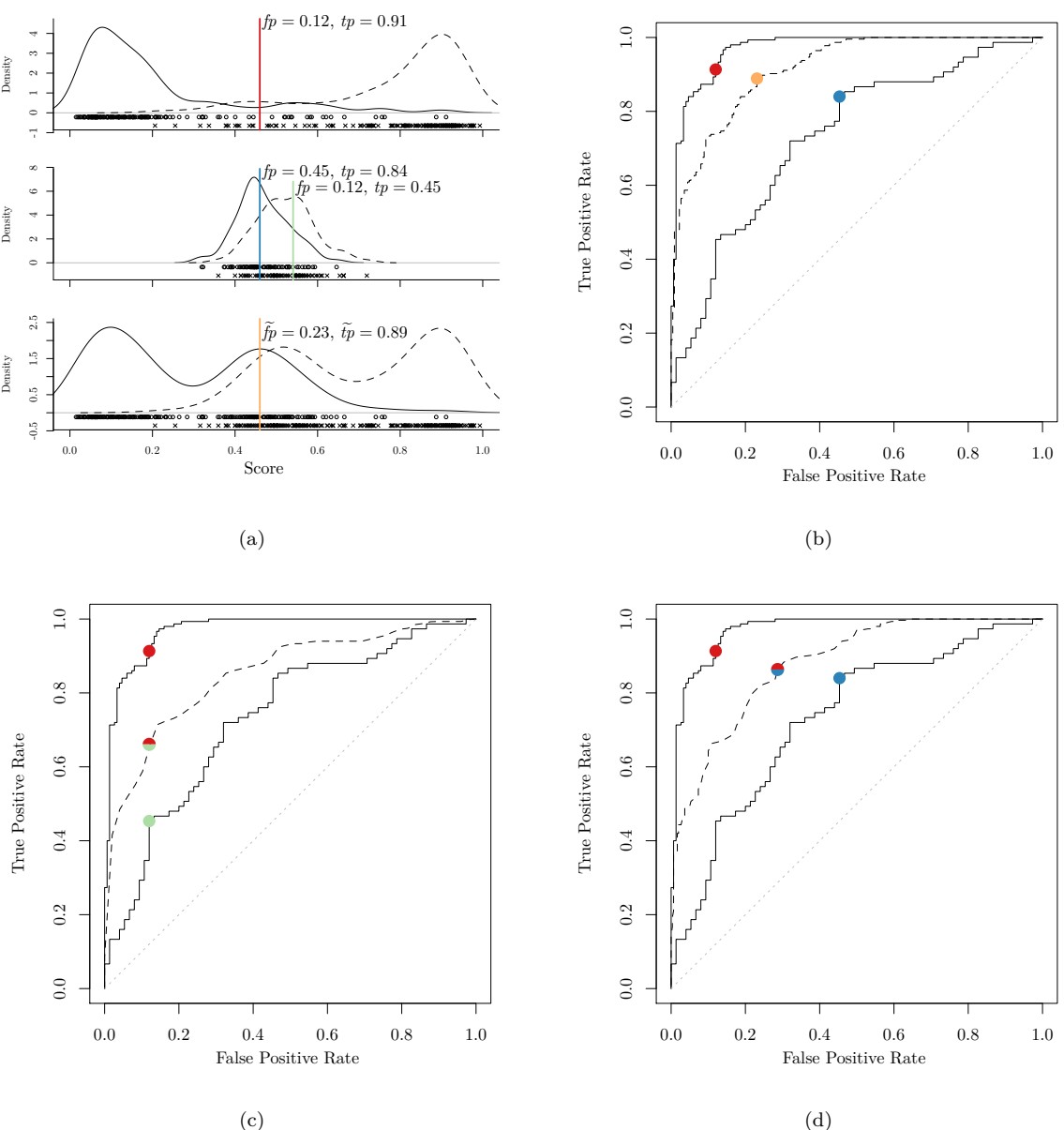

Figure 1: Illustration of three methods of averaging ROC curves. (a) Simulated classification scores for two datasets (top and middle) with density estimates of the negative and positive score distributions. The bottom plot shows density estimates of the pooled score distributions. (b) ROC curve pooling. (c) Vertical averaging. (d) Threshold averaging. In (b)–(d), coloured markers correspond to the fixed thresholds in (a); a split-colour marker indicates an average of the rates at the corresponding thresholds.

yielding the same false positive rate for the poorly-separable dataset is shown in green. The corresponding true positive values for each dataset are shown on the ROC curves in Figure 1c, along with the vertical average curve.

### 3.3 Threshold Averaging

Instead of sampling points based on their positions in ROC space, *threshold averaging* (Fawcett, 2006) samples uniformly from the threshold values $t$ and averages separately the *fp* and *tp* rates achieved by each

ROC curve for that value of $t$.

$$\overline{tp} = \frac{1}{M} \sum_{i=1}^{M} tp_i \qquad\qquad \overline{fp} = \frac{1}{M} \sum_{i=1}^{M} fp_i$$

$$= \frac{1}{M} \sum_{i=1}^{M} \frac{n_{P_i(t)}}{n_{P_i}} \qquad\qquad = \frac{1}{M} \sum_{i=1}^{M} \frac{n_{N_i(t)}}{n_{N_i}}.$$

The threshold average ROC curve is shown in Figure 1d. Note that for the fixed threshold shown in Figure 1a, the average $fp$ and $tp$ values do not equal $\widetilde{fp}$ and $\widetilde{tp}$ (the pooled rates). As $n_{N_1} > n_{N_2}$ and $n_{P_1} > n_{P_2}$, the well separable scores dominate in the calculation of $\widetilde{fp}$ and $\widetilde{tp}$, so the pooled ROC curve is drawn closer to the well separable ROC curve. If instead $n_{N_1} = n_{N_2}$ and $n_{P_1} = n_{P_2}$, then the threshold average curve and the pooled curve would in fact be identical.

## 4 Interpreting Average ROC Curves

In most cases, the methods described each result in different average ROC curves. This can be seen clearly in the simple illustration provided in Figure 1. Furthermore, the interpretation of each is subtly different. It is important therefore to consider which method is most appropriate for a given study.

*Pooling* classification scores disregards datasets. The pooled ROC curve can be considered as a weighted average of the individual ROC curves, weighted by the number of instances scored in each. Pooling produces only a single curve and so provides no information regarding uncertainty. If one dataset contains a substantially larger number of instances than the other datasets, the pooled ROC curve will be biased towards the ROC curve for that dataset. This should not be an issue in $k$-fold cross-validation, as the test sets are typically the same size. However, the pooling strategy—and equivalently, *threshold averaging*—assumes that the classifier outputs across folds of cross-validation are comparable and thus can be globally ordered. Parker et al. (2007) show that this assumption is generally not valid and can lead to large pessimistic biases in the estimation of the AUC and the shape of the ROC curve. Such biases are not suffered by *vertical averaging* (or its variants, discussed in Section 4.1); Chen & Samuelson (2014) prove that the AUC of the average curve equals the average of the individual AUCs.

In Fawcett (2006, p. 869), it is claimed that vertical averaging "is appropriate when the [false positive] rate can indeed be fixed by the researcher, or when a single-dimensional measure of variation is desired." Threshold averaging is presented as an alternative approach for situations when the false positive rate is not under the direct control of the researcher. The proposed solution is to average ROC points with respect to fixed thresholds, as these can be controlled by the researcher. We argue that this guidance is highly misleading, and the choice of averaging technique should instead be made under careful consideration of the *interpretation* of the resulting ROC curve. The vertical average ROC curve should be interpreted as the average true positive rate achieved amongst datasets for each fixed false positive rate, *allowing the threshold yielding that false positive rate to vary between datasets*. The threshold average ROC curve's interpretation is the average performance achieved amongst datasets *if the threshold is fixed across all datasets*.

The way in which the classifier is to be deployed should therefore be the first consideration to guide whether ROC curves should be averaged by threshold or averaged in ROC space. If a fixed threshold is used in practice, then threshold averaging is appropriate. Otherwise, averaging should take place in ROC space and the researcher must next consider the characteristics of the curve upon which evaluation or comparisons will be made.

### 4.1 ROC Space Averaging

In situations where we do not assume a fixed threshold, vertical averaging is preferable to threshold averaging to provide a representation of the average performance achieved. Indeed, in many operational settings, the researcher is concerned with the average true positive rate achieved for small false positive rates. However, in other settings it may be more relevant to assess the average false positive rate suffered in order to achieve a high true positive rate, say. This information will not be conveyed by the vertical average curve. Similarly,

it is common in biometrics studies (Wayman, 1999) to consider a ROC curve in terms of the ratio of the false acceptance rate ($fp$) and the false rejection rate ($1 - tp$). The equal error rate (EER), the point at which these rates are equal, is then often used to assess and compare classifier performance. The vertical average ROC curve will again be a misleading description of the average performance when considered from this point of reference.

When averaging ROC curves in ROC space, the researcher must consider how the curve will be read, i.e. the axis of reference from which characteristics of interest are compared. To compare performance with respect to fixed false positive rates, averaging should take place *vertically*; for performance with respect to fixed true positive rates, averaging should take place *horizontally*; for performance with respect to fixed error rate ratios, averaging should take place *diagonally*.

All three approaches can be generalised into a single procedure for performing averaging in ROC space (Chen & Samuelson, 2014). The $fp$ and $tp$ axes are first rotated clockwise by an angle $\theta$ by applying a rotation matrix

$$\begin{bmatrix} fp' \\ tp' \end{bmatrix} = \begin{bmatrix} \cos\theta & \sin\theta \\ -\sin\theta & \cos\theta \end{bmatrix} \begin{bmatrix} fp \\ tp \end{bmatrix}.$$

The procedure is then the same as vertical averaging: values of $fp'$ are sampled uniformly and the corresponding $tp'$ values are averaged, yielding $\overline{tp'}$. The $(fp', \overline{tp'})$ pairs are then rotated anti-clockwise to the original ROC space:

$$\begin{bmatrix} fp \\ \overline{tp} \end{bmatrix} = \begin{bmatrix} \cos\theta & -\sin\theta \\ \sin\theta & \cos\theta \end{bmatrix} \begin{bmatrix} fp' \\ \overline{tp'} \end{bmatrix}.$$

Vertical, horizontal and diagonal averaging can then be achieved using $\theta = 0$, $\pi/2$, $\pi/4$ respectively.

Averages along other directions may also be of interest. For example, denote the *cost* of misclassifying an instance from P and N by $c_P$ and $c_N$, respectively. If the ratio of these costs is known or estimated, the threshold that minimises cost corresponds to the point of intersection of the ROC curve and a line originating from the point (0,1) with slope $-\pi_P c_P / \pi_N c_N$, where $\pi_P$ and $\pi_N$ are the probability that an instance comes from $P$ and $N$, respectively (Adams & Hand, 1999). Hence, ROC curves can also be averaged with respect to fixed costs using $\theta = \arctan(\pi_N c_N / \pi_P c_P)$. However, Adams & Hand (1999) point out that it is rare in practice for these costs to be known precisely, hence the pre-eminence of ROC curves in practical applications.

### 4.2 Examples Revisited

It should now be clear that two fundamental considerations—the operational intention and the evaluation objective—should inform the choice of averaging method. The primary consideration relates to whether a classifier is intended to operate on all datasets using a common, fixed threshold. This dictates whether averaging should take place based on such fixed thresholds (i.e. pooling or threshold averaging) or instead in ROC space. If ROC space averaging is appropriate, a further consideration guides the choice of direction along which ROC points should be averaged. An average ROC curve can convey the performance of a classifier with respect to fixed false positive rates, true positive rates, error rate ratios or misclassification cost ratios. Crucially, the quantity remaining fixed should be informed by the objectives of the evaluation and should be clearly stated in the presentation of results.

We now reconsider the examples given in Section 1 and suggest appropriate solutions:

- **Comparing classifiers:** For each of $M$ devices in the enterprise network, ROC curves are constructed using the predictions make by both classifiers. The size of the test sets varies between devices, so pooling would bias an average ROC curve towards more active computers. For operational reasons, the team would like to be able to tune a single threshold for the anomaly detectors across all devices. The two algorithms should therefore be compared based on the threshold average ROC curve. A useful anomaly detector should produce few false alarms; therefore, the shape of the two curves in the bottom left corner of ROC space is of particular interest.

- **Comparing datasets:** For a given imaging device, each radiologist produces a set of scores based on their confidence level (on a scale of 1–10) of whether or not a tumour is present in each image.

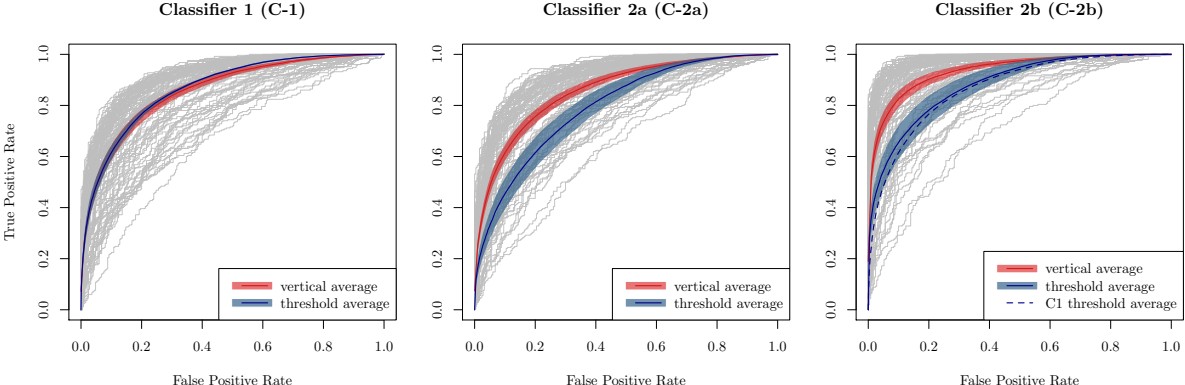

Figure 2: Simulated ROC curves for three classifiers and their averages under vertical and threshold averaging. The locations of the distributions used to simulate classification scores for C-1 are perturbed such that, for C-2a, the ROC curves—and hence the vertical average curves—are comparable but the threshold average curves differ, and for C-2b, the threshold average curves are comparable but the vertical average curves differ.

> Importantly, these scores are not calibrated probabilities, and the distribution of scores will vary between radiologists. For example, one radiologist may consistently make predictions with high confidence, i.e. 1s and 9s, while another might more often predict 4s and 6s; their discriminative power may nevertheless be the same. Pooling and threshold averaging, which assume a common threshold, do not make sense; averaging should take place in ROC space. The cost of missing a tumour is much greater than the cost of misdiagnosing one. Therefore, for a given imaging device, the threshold used in practice by a radiologist is likely to be selected to achieve some acceptably high true positive rate—which may not be known at the time of the study. To reflect this in the presentation of results, the researchers choose to compare the average false positive rate suffered for a range of fixed true positive rates. Horizontal averaging is appropriate.

- **Comparing hyperparameters:** For the reasons given in Section 4, threshold averaging is unlikely to be suitable for averaging over separate folds. To average in ROC space, the analyst must decide from what axis of reference the curves will be compared. If the cost of missing a churned customer is high, then horizontal averaging may be appropriate. However, the cost of false positives may also be high due to offering discounts to customers that were not planning on leaving. In this case, vertical averaging should be considered.

## 5 Simulated Illustration

We now present an example of a realistic use-case of ROC curve averaging in which the resulting conclusion depends on the method of averaging used. Suppose a researcher wishes to compare the performance of two different classifiers on data arising from a collection of $M$ datasets. Classification rules are learnt and scores generated by both classifiers using data from each dataset separately. The aim is to choose the classifier that has the better overall or average performance. We illustrate two scenarios.

**Scenario 1.** The output classification scores for two arbitrary classifiers were simulated as follows: For classifier 1 (C-1), negative and positive class scores for dataset $i$ are simulated from Gaussian distributions $N(\mu_{i0}, \sigma_{i0})$ and $N(\mu_{i1}, \sigma_{i1})$ respectively. For classifier 2a (C-2a), the scores for dataset $i$ are simulated from $N(\mu_{i0} + \varepsilon_i, \sigma_{i0})$ and $N(\mu_{i1} + \varepsilon_i, \sigma_{i1})$, where $\varepsilon_i \sim N(0, 1)$ is some random noise that differs between datasets but is the same for positive and negative populations within a dataset. As such, the locations of the modes of the score distributions of the two classifiers will be shifted but the class separability will be the same.

The first two panels of Figure 2 show ROC curves for both classifiers constructed using scores for $M = 100$ datasets simulated in this way. The vertical and threshold average ROC curves are shown for both classifiers. 95% confidence intervals for the averages were constructed at each point using the assumption of normality. Note that the vertical average curves remain approximately equal for both classifiers but the threshold average curves differ. This highlights the importance of considering the interpretations of average ROC curves in order to avoid erroneous comparisons. Here, if the operational intention is to fix a threshold across all datasets, but vertical averaging is used, no difference between the classifiers is evident and the researcher will choose either.

**Scenario 2.**   The opposite scenario is also possible. We simulate scores for an alternative classifier (C-2b), this time shifting the modes of C-1's score distributions unequally in order to increase the class separability. We do this by simulating negative class scores for dataset $i$ from $N(\mu_{i0} + \varepsilon_i, \sigma_{i0})$ as we did for C-2a but simulating positive class scores from $N(\mu_{i1} + |\varepsilon_i|, \sigma_{i1})$. For datasets where $\varepsilon_i < 0$, the score distributions become more separable. In Figure 2, the vertical average ROC curve for C-2b dominates both average curves for C-1; however the threshold average ROC curve remains approximately equal. Again, an averaging method unsuitable for the operational intentions can result in a misguided choice of classifier.

## 6   Conclusion

The naïve use of ROC curve combination methods is fraught with risk. We have summarised the common methods and pointed towards their interpretation in the operational context. Fundamentally, the appropriate choice of combination method *must* correspond to the operational approach for the classifier. To drive the message home, we provide a very simple illustration to show that incorrect decisions can result from ill-considered choice of combination method. We hope to have highlighted that in the context of classifier design and development, valuable information may be lost if due care and consideration is not paid when presenting and assessing results.

### Acknowledgments

The work of Jack Hogan was fully supported by a scholarship from QinetiQ. The authors are grateful to Professor David Hand and Dr Christoforos Anagnostopoulos for insightful comments on this manuscript.

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
