# OpenReview forum: "On Averaging ROC Curves"
_TMLR — Accepted by TMLR_

### Review · Reviewer_x8r3 · 2023-02-13

**Summary Of Contributions:**

This submission is a note that nicely summarizes different methods of averaging ROC curves. It also considers synthetic example settings where vertical and threshold averaging yield different conclusions.

**Audience:**

No

**Claims And Evidence:**

Yes

**Requested Changes:**

I do not believe that TMLR publishes tutorials, but this assumption may be wrong. There do not seem to be any original findings in the paper.

Other comments:

- "average true positive rate" - average over what? please specify this in the text

- Horizontal, vertical, and threshold averaging are also all available in:

Sing, T., Sander, O., Beerenwinkel, N., & Lengauer, T. (2015). Package ‘ROCR’. Visualizing the performance of scoring classifiers, 1-14.

and it would be useful to cite this paper and papers for other relevant software packages.


**Strengths And Weaknesses:**

The strongest aspect of the submission is the very clear exposition and the provided illustrations, based on synthetic examples, showing that it is important to choose an averaging method that is appropriate for the task at hand: using the incorrect method can lead to incorrect conclusions regarding the relative performance of two classifiers across a set of datasets.

In my view, this submission can best be classified as a tutorial on the appropriate use of averaging methods for ROC curves. The only components of the paper that is truly original (to my knowledge) are the synthetic example problems used to illustrate the different behaviors of two of the averaging methods.

Note that the generalization of averaging in Section 4.1 is in fact NOT novel. Rotating curves in order to average them has been proposed in

Chen, W., & Samuelson, F. W. (2014). The average receiver operating characteristic curve in multireader multicase imaging studies. The British journal of radiology, 87(1040), 20140016.

This paper is not cited in the submission.

---

> ### Author Response · Authors · 2023-02-24
> **Response to Reviewer x8r3**
>
> We thank the reviewer for their helpful comments, which we address below:
> * “Average true positive rate” — this is how we described an intuitive interpretation of the AUC. We agree that this statement is unclear. We will change the text to specify that the interpretation is the average true positive rate, *regarding all values of the false positive rate as equally likely* [1].
> * Package 'ROCR' — referencing relevant software packages is a good suggestion, which we will happily do. It is interesting to note that the package documentation and accompanying Bioinformatics paper [2] give no references to papers describing each of the methods or their interpretations.
> * Rotating curves — thank you for bringing this paper to our attention, which will be sure to cite. We used a rotation matrix in Section 4.1 in order to illustrate that curves can be averaged along an arbitrary direction, and gave an example (the case of known misclassification costs) of why one may wish to do so. It was not our intention to claim a novel method of averaging ROC curves but rather to emphasise the difference between averaging curves along some direction of interest in 'ROC space' and averaging curves with respect to fixed thresholds. We are happy to modify the text to make this more clear.
>
> [1] Hand, David J. "Measuring classifier performance: A coherent alternative to the area under the ROC curve." *Machine Learning* 77.1 (2009): 103-123. \
> [2] Sing, Tobias, *et al*. "ROCR: Visualizing classifier performance in R." *Bioinformatics* 21.20 (2005): 3940-3941.

---

### Review · Reviewer_AvaT · 2023-02-22

**Summary Of Contributions:**

The paper studies various ways how ROC curves obtained from different datasets can be averaged and what the implications are for each kind of averaging. It gives a toy example of how choosing the wrong method can lead to wrong decisions.

**Audience:**

Yes

**Broader Impact Concerns:**

No concerns.

**Claims And Evidence:**

No

**Requested Changes:**

Describe scenarios very clearly (what is known, what is not known, what is the ultimate measure of success) and then describe the solutions for these scenarios. See the above explanations regarding the problems with the current text.

Perhaps address the cross-validation scenario separately and in more detail.


**Strengths And Weaknesses:**

Strengths:
(S1) Mathematical notation and formulas seem to be without errors.

(S2) The text is understandable (but not sufficient, see comments below).

Weaknesses:
(W1) Several questions have not been addressed sufficiently in the paper:

* In which situations does it make sense to perform averaging of ROC curves across multiple datasets? More particularly, what is the final decision that needs to be made based on the averaged ROC curve(s)? Is it classifier selection? If yes, then what is the selection criterion that needs to be optimised for? To me it seems that maintaining all ROC curves and making separate decisions on each dataset is a method which is more likely to be successful than making decisions based on the averaged ROC curve. Perhaps cross-validation as a use-case (mentioned in the introduction) indeed might justify estimating one aggregated ROC curve, but this situation would then need a dedicated analysis, which has not been provided in this study. It is possible that authors had mostly different examples in mind, but I think that averaging of ROC curves was not justified sufficiently for these examples. The paper touches on this only very briefly, e.g. Section 3 when introducing averaging of ROC curves states that 'Often we have multiple independent datasets of this type and a set of classification scores for each' and 'for the reasons given in Section 2.3, it may be preferable to construct a single ROC curve that serves as a summary'. Section 2.3 discusses the importance of ROC curves in general, but not importance of averaging ROC curves. Introduction states that 'It is perhaps natural in these settings to seek to report a single ROC curve summarising the performance of all the individual classifiers. Such a curve will be useful for classifier development and selection.'  In my opinion, these statements do not provide sufficient justifications and explanations about when it make sense to perform averaging of ROC curves across multiple datasets.

* When to use which averaging method? In a scenario that requires the threshold to be the same fixed quantity across all datasets, it becomes quite clear indeed, that one should aggregate ROC curves also by threshold. But when to use pooling and when threshold averaging? And if threshold does not have to be the same, then which averaging method to use? There is a continuous family of averaging at any angle but there are no clear instructions about which method to use.

I understand that it is hard to address all the concerns above because it is really the domain that needs to inform the choice of evaluation methodology. However, the paper should at least then identify some scenarios very clearly and provide methods to address these scenarios.

(W2) It is not clear what the novelty of the paper is. The averaging methods are not new (except for averaging at any angle, which seems a bit exotic). The papers proposing these methods originally, have already covered some of the similar discussions as this paper, particularly the papers by David J. Hand. The toy example does not seem to add very much to the discussion.

---

> ### Author Response · Authors · 2023-02-25
> **Response to Reviewer AvaT**
>
> We thank the reviewer for their helpful comments, which we address below:
> * W1 — As suggested, there are a wide variety of application domains and evaluation decision processes in which average ROC curves are used; e.g. comparing different algorithms (classifier selection), comparing different measurement devices (MRMC studies), or comparing different classifier hyperparameters (cross validation). Indeed, this contributes to the importance of clear guidance on correct usage. If a practitioner is used to employing a certain method for one decision process (say, classifier selection), they may erroneously use the same method for a different decision process (say, cross-validation). By discussing the topic of averaging ROC curves in complete generality, we may have lost clarity. We are happy to include a selection of explicit example use-cases for average ROC curves, highlighting the decision process being considered in each case.
> * W2 — The purpose of the paper is not to propose a novel averaging methodology but rather to provide for the first time a clear and dedicated treatment of average ROC curves. We believe this will be a valuable resource for the machine learning community.

---

> > ### Comment · Reviewer_AvaT · 2023-02-27
> > **Re: Response to Reviewer AvaT**
> >
> > W1:
> >
> > Thanks for your clarification. Please consider my statement 'maintaining all ROC curves and making separate decisions on each dataset is a method which is more likely to be successful than making decisions based on the averaged ROC curve' and list situations where this statement is likely to be wrong.
> >
> > W2:
> >
> > I agree that the goal is not to propose a novel averaging methodology, but I am struggling to see the paper as 'for the first time a clear and dedicated treatment of average ROC curves'. Please highlight elements of the treatment that have not been covered by the existing literature. One element is averaging at any angle, but which are other elements or insights that have not been covered before?

---

### Review · Reviewer_uaRw · 2023-03-07

**Summary Of Contributions:**

The paper discusses how ROC curves issued from multiple data sets/classifiers should be aggregated together according to the user needs. Some simulated experiments emphasize the differences between the methods, and their respective interest is discussed. A specific attention is given to the case of vertical averaging (averaging over a fixed false positive rate), for which an extension is given based on the idea that the value of interest to be fixed may vary accross applications (e.g., fixing the ratio between true and false positive rates, or fixing the true positive rate).

**Audience:**

Yes

**Broader Impact Concerns:**

No specific concerns

**Claims And Evidence:**

Yes

**Requested Changes:**

I am quite satisfied with the current state of the contribution. The only thing on which I would like more precision is what authors exactly mean by commensurate distributions at the start of Section 4.

**Strengths And Weaknesses:**

Overall I found the contribution to be well-written, clear and rather convincing in its argument. I also think that it fits quite well with TMLR format, in the sense that it presents a short, not too dense yet interesting contribution, and is written so that it maximizes accessibility to the readership.

As I am not a specialist of ROC curves, there may be subtleties I did not catch, but I thank the authors for making the paper really clear and accessible.

---

### Author Response · Authors · 2023-03-15
**Summary of Revised Manuscript**

We would like to thank the reviewers for their comments and constructive suggestions. We have revised the paper to address any concerns that were raised.
Here we summarise the major changes:
* We expand the introduction to provide considerably more context as to why ROC curves are useful, why an average ROC curve is useful and why the existing literature is lacking. Specifically:
  * We emphasise the advantage of ROC curves over scalar measures of performance, particularly in situations where misclassification costs and class distributions are unknown or imprecise.
  * To help reinforce the message of the paper, we describe three hypothetical examples in which an average ROC curve may be used in experimentation. This helps to build a narrative and convey the idea that there is a variety of situations in which average ROC curves may be used in practice, which may be quite different in nature. These are revisited later in the paper.
  * We highlight the lack of guidance in the existing literature, including two examples of this fact being explicitly noted by others.
  * We reference software packages that provide ROC curve averaging functionality.
* In Section 2.3, we correct wording, which clarifies the interpretation of the AUC.
* In Section 3, we reiterate why a scalar measure of performance may not be sufficient for a given study; hence, the average of many scalar measures may also be insufficient.
* In Section 4, we pay specific attention to k-fold cross-validation and provide references that argue that threshold averaging and pooling may not be appropriate in this context. Our previous reference to ‘commensurate distributions’ is clarified here.
* We cite the Chen & Samuelson paper when describing the procedure for carrying out ROC space averaging.
* We added a new section, Subsection 4.2, which revisits the three hypothetical examples that were introduced at the beginning of the paper. Having explained the various approaches to averaging and their respective interpretations, we provide specific guidance on correct usage in these three situations.

We thank the reviewers again for their helpful suggestions. We would be happy to address any additional questions and concerns the reviewers might have.

---

### Decision · Action_Editors · 2023-04-30

**Recommendation:** Accept with minor revision

**Comment:**

The authors study different alternatives of averaging the Receiver Operating Characteristic (ROC) curves, and point out that different averaging approaches mean different interpretations and can lead to different scientific conclusions.

The reviewers agree that the paper provides a good survey for aggregating ROC curves, and a clear and convincing guidance for practitioners. One reviewer pointed out that Section 4.2 needs further clarification. The authors are expected to enhance the rigor in the arguments to support the claims and differentiate horizontal/vertical averaging more clearly.


**Audience:**

Yes

**Claims And Evidence:**

Yes (except that Section 4.2 needs a bit of fixing)

---

> ### Author Response · Authors · 2023-05-26
> **Camera-ready submission**
>
> Thank you for accepting our manuscript for publication. We have uploaded a camera-ready version, including further clarification to Section 4.2. We would like to thank also the reviewers for taking the time to provide helpful feedback and suggestions, which have improved the quality of the final paper.